# IPS (In-Plant System) Delivery of Double-Stranded *Vitellogenin* and *Vitellogenin receptor* via Hydroponics for Pest Control in *Diaphorina citri* Kuwayama (Hemiptera: Psyllidae)

**DOI:** 10.3390/ijms24119497

**Published:** 2023-05-30

**Authors:** Hailin Li, Junlan Mo, Xiaoyun Wang, Biqiong Pan, Shu Xu, Shuangrong Li, Xialin Zheng, Wen Lu

**Affiliations:** 1Institute of Plant Protection, Guangdong Academy of Agricultural Sciences, Key Laboratory of Green Prevention and Control on Fruits and Vegetables in South China Ministry of Agriculture and Rural Affairs, Guangdong Provincial Key Laboratory of High Technology for Plant Protection, Guangzhou 510640, China; 2Guangxi Key Laboratory of Agric-Environment and Agric-Product Safety, College of Agriculture, Guangxi University, Nanning 530004, China

**Keywords:** *Diaphorina citri*, RNAi, vitellogenin, egg formation, integrated pest management

## Abstract

*Diaphorina citri*, a vector of citrus huanglongbing (HLB) disease, frequently leads to HLB outbreaks and reduces Rutaceae crop production. Recent studies have investigated the effects of RNA interference (RNAi) targeting the *Vitellogenin* (*Vg4*) and *Vitellogenin receptor* (*VgR*) genes, which are involved in egg formation in this pest, providing a theoretical foundation for developing new strategies to manage *D. citri* populations. This study presents RNAi methods for *Vg4* and *VgR* gene expression interference and reveals that ds*VgR* is more effective than ds*Vg4* against *D. citri*. We demonstrated that ds*Vg4* and ds*VgR* persisted for 3–6 days in *Murraya odorifera* shoots when delivered via the in-plant system (IPS) and effectively interfered with *Vg4* and *VgR* gene expression. Following *Vg4* and *VgR* gene expression interference, egg length and width in the interference group were significantly smaller than those in the negative control group during the 10–30-day development stages. Additionally, the proportion of mature ovarian eggs in the interference group was significantly lower than that in the negative control group at the 10, 15, 20, 25, and 30-day developmental stages. Ds*VgR* notably suppresses oviposition in *D. citri*, with fecundity decreasing by 60–70%. These results provide a theoretical basis for controlling *D. citri* using RNAi to mitigate the spread of HLB disease.

## 1. Introduction

*Diaphorina citri* (Kuwayama) (Hemiptera: Psyllidae), a vector of citrus huanglongbing (HLB), frequently causes disease outbreaks [1,2]. Research has shown that *D. citri* adults can disperse over long distances under high wind speeds in autumn, and the widespread cultivation of Rutaceae crops globally has exacerbated the growth of individual *D. citri* populations and the resulting damage [3,4]. Consequently, controlling the *D. citri* population is crucial for suppressing the spread of HLB disease [5,6,7]. Currently, many orchards predominantly rely on pesticides for *D. citri* management. However, the long-term use of chemical pesticides increases the proportion of insecticide-resistant individuals within the *D. citri* population, leading to elevated control costs. Hence, it is essential to explore new methods for the integrated management of *D. citri*.

Advancements in molecular biology technology have enabled the application of sophisticated genetic engineering techniques, such as RNA interference (RNAi) and CRISPR-associated protein 9 (CRISPR/Cas9), in pest management. Recent studies have focused on the interference and knockout of oviposition genes closely associated with pest population development, including the *phospholipase A2*, *sugar gustatory receptor*, *odorant receptor coreceptor*, *diapause hormone*, *dopa decarboxylase*, and *sucrose hydrolase gene families* [8,9,10,11,12,13]. The *Vitellogenin* (*Vg*) and *Vitellogenin receptor* (*VgR*) gene families have been shown to play a direct role in egg formation [14]. For instance, knocking out the *Tudor* gene in *Bactrocera dorsalis* (Diptera: Trypetidae) resulted in reduced ovarian development and mating rates, with ds*Tudor* interference efficiency reaching 95%; interfering with the expression of *Vg* and *VgR* genes effectively hindered *Vitellogenin* accumulation and mature egg formation in the ovaries [15,16,17]. A significant reduction in the next generation’s population size will occur if the pest cannot accumulate sufficient *Vitellogenin*.

It is well established that *Vg* protein is critical to insect egg production and embryonic development. In order to mitigate the damage caused by *D. citri* and suppress the spread of HLB disease, RNAi has been employed to inhibit *Vg* and *VgR* gene expression in the ovaries, thus controlling egg formation and representing a novel approach to managing *D. citri* populations. A study revealed that female egg production requires the mobilization of nutrients for egg cells in the ovary to synthesize *Vitellogenin* [18]. *Vitellogenin* is produced in the fat body, secreted into the hemolymph, and subsequently incorporated into developing oocytes via *VgR*-mediated endocytosis [14]. Consequently, RNAi was utilized to interfere with the expression of female *Vg* and *VgR* genes and egg formation as a viable method for controlling pest damage. Previous research indicated that *Vg* gene expression can directly or indirectly affect insect ovary development, while *VgR* gene expression can influence egg formation, oviposition behavior, and other physiological activities related to oviposition [15,18].

*Vitellogenin-1-like-1* (*Vg1*) (Gene id: LOC103,523,873; gene full length: 1858 bp), *Vitellogenin-1-like-2* (*Vg2*) (Gene id: LOC103,523,874; gene full length: 1116 bp), *Vitellogenin-2-like* (*Vg3*) (Gene id: LOC103,513,507; gene full length: 1197 bp), *Vitellogenin-3-like* (*Vg4*) (Gene id: LOC103,523,199; gene full length: 4808 bp), *Vitellogenin-like* (*Vg5*) (Gene id: LOC113,469,177; gene full length: 1002 bp), and *VgR* (Gene id: LOC103,524,089; gene full length: 5950 bp) were identified. Bioinformatics information was comprehensively described based on the *D. citri* female transcriptome and proteome we previously reported (NCBI: PRJNA669507; ProteomeXchange: PDX022359) [19]. However, the developmental characteristics of the ovary were not documented after interference with the *Vg* and *VgR* gene expression in *D. citri*. Currently, RNAi has been widely applied in insect gene function studies, primarily employing the delivery of double-stranded ribonucleic acid (dsRNA) into insects via quantitative injection. The advantage of the quantitative injection method is the control over the amount of dsRNA administered to insects. However, its disadvantages include causing injury to insects, resulting in a reduced lifespan of research subjects and the inability to perform multiple injections. Small insects such as *D. citri* and aphids are not suitable for using the injection method to deliver dsRNA. Instead, the direct oral feeding of dsRNA on *D. citri* was mainly applied to verify the effect or function of relative genes for a period of time [20,21]. The indirect delivery of dsRNA to *D. citri* through plants was less reported and the observation time was usually short, with the longest observation time not exceeding 25 days [22]. The effects of dsRNA long-term (more than 30 days) feeding on the gene interference expression of *D. citri* are almost unknown.

We further discovered that the Vg4 protein, corresponding to the *Vg4* gene, had the most complex amino acid structure in the *Vg* gene family and was highly expressed in the female abdomen. This finding suggests that the *Vg4* gene may play a crucial role in egg formation physiology. There was only one *VgR* gene, which was directly selected as an interference target. In this study, we improved and innovated Andrade’s method of feeding dsRNA through plants (in-plant system: IPS) to observe the long-term effects of feeding ds*Vg4* and ds*VgR* on the oviposition activities of *D. citri* [22]. We tested whether ds*Vg4* and ds*VgR* could effectively reduce the expression levels of *Vg4* and *VgR* genes in *D. citri*, thereby interfering with egg formation. To address our research question, we established two treatments (i.e., ds*Vg4* and ds*VgR*) and two controls (i.e., double-stranded green fluorescent protein (ds*GFP*) and blank control) to examine the following aspects: (1) the stability of dsRNA in *Murraya odorifera* shoots using IPS delivery; (2) *Vg4* and *VgR* gene expression levels and ovarian development; (3) differences in egg formation and the number of malformed eggs and nymphs; (4) variations in fecundity and egg hatchability; and (5) suppression effects of ds*Vg4* and ds*VgR* on female oviposition.

## 2. Results

### 2.1. Stability of dsRNA in M. odorifera Shoots

Gel electrophoresis was employed to assess the stability of ds*Vg4*, ds*VgR*, and ds*GFP* in treated *M. odorifera* shoots over six days. Clear gel electrophoresis bands were observed at 1–6 days in the *M. odorifera* shoots treated with ds*GFP*, indicating the feasibility of IPS for delivering dsRNA (Figure 1). Ds*Vg4* and ds*VgR* displayed clear bands at 1–2 days, suggesting that ds*Vg4* and ds*VgR* maintained a relatively complete double-stranded structure during this period. Once *D. citri* females fed on the *M. odorifera* shoots for 1–2 days, ds*Vg4* and ds*VgR* readily entered the female body. The bands became blurred after 3–6 days, signifying that ds*Vg4* and ds*VgR* started to decompose at this stage. However, traces of bands in ds*Vg4* gel electrophoresis indicated that the structures of some ds*Vg4* and ds*VgR* remained intact at 3–6 days (Figure 1).

Chromas analysis of gene sequencing results for ds*Vg4* and ds*VgR* was consistent with the template gene used to synthesize dsRNA (Appendix A). Gel electrophoresis was also performed on the ds*GFP* group and blank control group using *Vg4* and *VgR* PCR primers to demonstrate the reliability of the PCR results. No clear bands were observed in the gel electrophoresis when *Vg4* and *VgR* PCR primers were used in the PCR process for the ds*GFP* group and blank control group, proving the specificity of *Vg4* and *VgR* PCR primers (Appendix A). The original gel electrophoresis photos of these PCR tests are available in the Appendix A.

### 2.2. Gene Expressions of Vg4 and VgR and Ovarian Development

After *D. citri* females were fed ds*Vg4*, the expression of the *Vg4* gene decreased during the 1–30 d period. The expression of the *Vg4* gene was most significantly decreased at 15 d, and the relative expression declined by 99.06% (*p <* 0.05). However, the expression of the *Vg4* gene abnormally increased at 20 d, with the relative expression rising by 143.65% (*p <* 0.05) (Figure 2). This finding indicates that, at the 20 d developmental stage, females exhibited the strong immune regulation of ds*Vg4*. By dissecting the internal reproductive organs of females with interference in *Vg4* gene expression, we observed that the number of eggs in the ovary of the *Vg4* interference group was lower than that in the negative control group at 1–30 d. Ovaries from females in the negative control group began to form a large number of eggs at 15 d, and a certain number of eggs could be observed in the ovaries at 15–30 d, which was similar to the normal development rate (Appendix A). Fewer mature and immature eggs were observed in the ovaries of the interference group, and a large number of eggs had not yet formed in the ovaries.

After *D. citri* females were fed ds*VgR*, the expression of the *VgR* gene decreased during the 1–30 d period. The expression of the *VgR* gene decreased most significantly at 25 d, with the relative expression declining by 99.57% (*p* < 0.05). The expression of the *VgR* gene abnormally increased at 20 d, and the relative expression rose by 256.93% (*p* < 0.05) (Figure 3). By dissecting the internal reproductive organs of females with interference in *VgR* gene expression, we observed that the number of eggs in the ovary of the interference group was lower than that of the negative control group at 1–30 d, similar to the findings for the *Vg4* interference group (Table 1).

### 2.3. Effects of dsVg4 and dsVgR on the Morphology of Eggs and Nymphs

Interference with *Vg4* and *VgR* gene expression in *D. citri* females resulted in a relatively small number of mature ovarian eggs at the 15–30 d stage (Figure 2 and Figure 3, Table 1). At the 5 d developmental stage, the ovaries of both the interference group and the negative control group had not formed any eggs. Mature eggs were only found in the dissected ovaries of the *Vg4* and *VgR* interference groups at the 10 d developmental stage. At this stage, eggs in the *Vg4* interference group exhibited an average length of 136.30 ± 15.55 µm, an average width of 62.58 ± 6.30 µm, a maximum number of mature eggs of 1.03 ± 0.37 eggs, and a maximum percentage of mature eggs of 8.33 ± 1.67%. In the *VgR* interference group at the 10 d developmental stage, eggs exhibited an average length of 123.32 ± 30.03 µm, average width of 61.62 ± 6.59 µm, maximum number of mature eggs of 0.75 ± 0.25 eggs, and maximum mature egg percentage of 7.75 ± 0.50%. The egg size and mature egg proportion of the *Vg4* and *VgR* interference groups were lower than those of the negative control group at the 10 d developmental stage. At the 15–30 d developmental stage, mature eggs were not observed in the *Vg4* and *VgR* interference groups, and only a few immature eggs were observed in the ovarioles.

After interfering with *Vg4* and *VgR* gene expression in *D. citri* females, abnormal eggs and nymphs were found on the *M. odorifera* shoots (Figure 4). Females in the negative control group produced normal eggs, which were shaped like droplets and had smooth, yellow surfaces. In contrast, the egg surfaces of the *Vg4* and *VgR* interference groups were abnormal, appearing shriveled and brown compared to normal eggs. Normal nymphs in the negative control group had smooth backs and yellow bodies, while abnormal nymphs in the *Vg4* and *VgR* interference groups exhibited wrinkled backs, brownish-yellow bodies, and some translucent nymph bodies in the *Vg4* interference group. None of the abnormal eggs hatched, and the abnormal nymphs displayed abnormal characteristics at the 1–2 instar stage.

### 2.4. Comparison of Oviposition Suppression between dsVg4 and dsVgR

In the *Vg4* interference group, a total of 1379 eggs were collected, with an average of 45.97 ± 7.31 eggs per female. Out of these, 1190 eggs hatched into nymphs, resulting in an average hatchability of 84.41 ± 6.50%. The total deformity rates for eggs and nymphs were 4.42% and 4.12%, respectively (Table 2). In the *VgR* interference group, a total of 638 eggs were collected, with an average of 20.80 ± 9.37 eggs per female. Among these, 594 eggs hatched into nymphs, yielding an average hatchability of 81.11 ± 7.30%. The total deformation rates for eggs and nymphs were 6.80% and 1.35%, respectively. In the ds*GFP* and blank control groups, 2804 and 2457 eggs were collected, with an average of 93.47 ± 10.20 and 84.97 ± 12.28 eggs per female, respectively. In these groups, 2570 and 2380 eggs hatched into nymphs, with average hatchability rates of 94.89 ± 2.76% and 96.67 ± 1.94%, respectively. No deformed eggs or nymphs were observed in these groups.

## 3. Discussion

Our findings indicate that RNAi through the IPS method could successfully suppress egg formation in *D. citri*. This suppression is attributed to the interference of ds*Vg4* and ds*VgR*, which disturbed ovarian development and subsequently decreased fecundity and egg hatchability [23,24,25]. This was especially true for ds*VgR*, which proved to be more effective than ds*Vg4* in terms of suppressing egg formation.

Based on the results of dsRNA stability in *M. odorifera* shoots, our study demonstrated that delivering dsRNA to an insect is feasible through plant absorption (Figure 1). Since the dsRNA solution could be directly analyzed by RT-PCR even after plant absorption, the total plant RNA could be extracted for RT-PCR to verify the presence of dsRNA (Appendix A). This detection method is widely used in research on dsRNA viruses and dsRNA transgenic crops [26,27,28]. DsRNA can persist for a period of time in liquid form under certain environmental conditions. For instance, a study found that after leaf tissue absorbed ds*GFP*, the tissue tested positive for ds*GFP* at 24 h post-treatment and remained positive for up to 40 days post-treatment [22]. DsRNA can be detected by reverse transcription reaction after being absorbed by plants, as dsRNA is not stable in the environment. It has been reported that the half-life of naked siRNA in serum ranges from several minutes to about an hour [26]. This indicates that dsRNA has difficulty maintaining the stability of its double-stranded structure for extended periods in the complex internal environment of plants. Consequently, if dsRNA persists in plants for more than 24 h and no suitable amplification site can be found, dsRNA will transition to a single strand and further decompose. The cDNA and PCR reagents can be used to detect the presence of dsRNA when the dsRNA in plants becomes single-stranded. It is worth noting that, compared to the *Vg4* and *VgR* interference treatment groups, *M. odorifera* shoots treated with ds*GFP* (726 bp) displayed relatively clear bands within 1–6 days, suggesting that the half-life of dsRNA may be related to the length of its base sequence (Figure 1 and Appendix A). A study found that the 1454 bp dsRNA virus of *Trichomonas vaginalis* requires boiling water treatment for qRT-PCR analysis, which also supports the idea that a longer dsRNA base sequence provides greater stability and a longer half-life [27]. This finding indicates that designing target gene dsRNA interference fragments and extending the base length of dsRNA within a controllable range may increase dsRNA’s half-life, although more experiments are needed for verification. In this study, the gel electrophoresis of ds*Vg4* and ds*VgR* demonstrated that ds*Vg4* and ds*VgR* could persist for 2–5 days (Figure 1), indicating that ds*Vg4* and ds*VgR* could be absorbed by *M. odorifera* shoots and accumulate in the leaves, effectively delivering ds*Vg4* and ds*VgR* to females feeding on *M. odorifera* shoots. Consequently, dsRNA formulated into an appropriate molecular dosage form may be suitable for the integrated management of *D. citri* populations and/or can be used in combination with certain chemical pesticides [29].

Gene expression data demonstrated that the *Vg4* and *VgR* expression levels were significantly affected by ds*Vg4* and ds*VgR* in *D. citri*. At the 1–30-day developmental stage, the relative expression of the *Vg4* and *VgR* genes was detected at six time points, revealing an overall decrease in their expression. However, the relative expression of the *VgR* gene abnormally increased by 256.93% at the 20-day developmental stage (Figure 3). This phenomenon may be attributed to cellular immune regulation by dsRNA. Previous studies have indicated that dsRNA-degrading enzymes (dsRNases) are important factors in reducing RNA interference efficiency in different insect species and can lead to an increase in gene expression [30,31].

The gene expression results show that after *D. citri* females were fed dsRNA for an extended period, they were likely to exhibit a set of immune physiological effects against dsRNA through an immune response, which may result in a decline in the interference effect of dsRNA on the target gene expression. Typically, the secretion of dsRNA-degrading enzymes is part of the insect immune response [32]. Whether the long-term feeding of dsRNA leads to sustained immune response activation remains unclear, and research on pest resistance to dsRNA pesticides is not comprehensive.

The large-scale application of dsRNA for managing *D. citri* would begin by bypassing the immune response physiology of *D. citri* and enhancing the integrated management effect of dsRNA without stimulating the immune response of *D. citri*. However, further research is required to understand the immune response in *D. citri* induced by dsRNA and the related mechanisms.

Ds*Vg4* and ds*VgR* significantly suppress egg formation in *D. citri*, demonstrating that the *Vg* and *VgR* gene families have a crucial impact on ovarian development [19]. This study’s results also revealed that the number of mature eggs in the ovary and the number of eggs/female significantly decreased after ds*Vg4* and *VgR* interference with the *Vg4* and *VgR* genes. The length and width of female eggs appeared intermittently in the 10–30-day *VgR* interference group (Table 1) because ds*VgR* disrupted the formation of the *VgR* receptor, preventing the normal transport of *Vg* proteins to the ovary [33].

Although the ovary could not accumulate enough *Vg* proteins to form eggs, ds*VgR* only interfered with, rather than blocked, *Vg* protein accumulation. Females could still carry out oviposition activities to deposit eggs on plants. After oviposition activities, the ovary needed to re-accumulate *Vg* protein in case of insufficient *VgR* receptor. Consequently, the length and width of eggs appeared intermittently in the 10–30 day *VgR* interference group. The suppressive effect of ds*VgR* on egg formation was stronger than that of ds*Vg4*, suggesting that the effect of *VgR* gene expression was more influential than the *Vg4* gene on mature egg formation [34]. However, the deletion of the *VgR* receptor can lead to the insufficient accumulation of *Vg4* protein in the ovary. Thus, after interference with *Vg4* and *VgR* gene expression, the ovarian development characteristics between the *Vg4* and *VgR* interference groups were similar.

From the experimental results, there may be an upstream and downstream relationship between the *Vg4* gene and *VgR* gene expression, with the *VgR* gene likely being the upstream gene of the *Vg4* gene. In reality, the gene expression network involved in ovarian *Vg* protein synthesis and transport may be more complex and requires a comprehensive and systematic study of the *Vg* protein synthesis and transport gene family. Constructing a gene bank of *D. citri* egg formation-related genes would be beneficial for the future integrated management of *D. citri* populations using molecular biology technology.

Although the RNAi of the expression of the *Vg4* and *VgR* genes could not entirely block female oviposition, if combined with other integrated pest management (IPM) control methods and strictly following the IPM principle, the occurrence rate of *D. citri* outbreaks could be reduced to very low levels. In this study, an interference system for the long-term feeding of ds*Vg4* and ds*VgR* to knock down the expression of female *Vg4* and *VgR* genes was established. It can continuously feed adults with dsRNA from the emergence to death stages, allowing for multiple oviposition phenotype and behavior studies. In the context of the successful RNAi of ds*Vg4* and ds*VgR*, the functions of more oviposition-related genes can be evaluated in *D. citri*, such as upstream and downstream gene families associated with *Vg4* and *VgR*.

The functions of regulatory genes of *Vg* and *VgR* have been reported in some insects, while relative reports in *D. citri* are limited. For example, interference of the *Krüppel homolog 1* (*Kr-h1*) gene in *Bombyx mori* female pupae severely inhibited *VgR* transcription, leading to reduced *Vg* deposition in oocytes [35]; Vg synthesis is directly affected by *doublesex* in *Agrotis ipsilon*, *mucin* genes in *Nilaparvata lugens* [36,37]; as well as *DOPA decarboxylase* in *Drosophila sechellia* (Diptera: Drosophilidae) and *Plutella xylostella* (Lepidoptera: Plutellidae) [38,39], etc. Therefore, further studies on the regulatory genes associated with the expression of the *Vg* and *VgR* gene family can help provide theoretical support for the systematic control of *D. citri* populations using RNAi. In addition to the results in the current study, further experiments are required to elucidate the impact of the *Vg4* and *VgR* gene families on the oviposition physiology and behavior of *D. citri* females. These research efforts can provide a gene data theoretical basis for controlling the damage caused by *D. citri* populations using RNAi technology in the future [40].

Moreover, in terms of pest management, the application of RNAi not only depends on the suppressive effect of dsRNA on oviposition but also requires the establishment of proper methods for utilizing dsRNA. To minimize insect resistance and the environmental impact of insecticides, it is crucial to develop appropriate molecular pesticide formulations with dsRNA. Previous studies have reported that the effectiveness of dsRNA combined with specific nanomaterials may be superior to some chemical pesticides against insect pests, such as *Anopheles gambiae*, *B. dorsalis*, and *Apolygus lucorum* [41,42,43,44,45,46,47]. If dsRNA can be combined with nanomaterials or formulated into corresponding agents, it can be employed for the large-scale integrated management of *D. citri* populations. However, it is also necessary to evaluate the interference effect of dsRNA on different day-ages in *D. citri* and determine the optimal time for suppressing female oviposition.

## 4. Materials and Methods

### 4.1. Insect Rearing and Collection

Adult *D. citri* were collected from *M. odorifera* plants in the citrus orchard of Guangxi University, China (108.290° E, 22.849° N), and raised in five indoor cages (90 × 90 × 100 cm^3^). *M. odorifera* plants with heights ranging from 80 to 100 cm were transferred to insect cages as food. Six hundred adults that emerged after 8–11 days were paired in the indoor cages mentioned above, and tender shoots of *M. odorifera* seedlings were used as oviposition substrates. Newly emerged adults of the next generation were used as the experimental insects. All experiments were conducted indoors at a temperature of 25 ± 1 °C, relative humidity of 75 ± 5%, and a light: dark cycle of 14:10 h.

Males and females that emerged on the same day were used in the following experiments, and all were reared in plastic containers. The plastic container consisted of a cup (148 mm high × 60 mm wide at the bottom), a cover (95 mm in diameter), and a plastic platform (70 mm in diameter) and contained 1–2 *M. odorifera* shoots and a 1.5 mL Eppendorff (ep) tube. A cover could be opened and closed to facilitate the replacement of dsRNA solution and *M. odorifera* shoots (Figure 5). For the RNAi assay, dsRNA was dissolved in water, added to the tube, and then transported to the tender shoot through the xylem of *M. odorifera* for feeding the females. Mating and oviposition among the adults in the rearing container were observed and recorded.

### 4.2. Ovary Anatomy and Measurement

Female *D. citri* were rendered immobile by dipping in liquid nitrogen prior to dissection and quickly transferred to alcohol-treated wax tablets for ovary dissection. The specimens were soaked in precooled 1X PBS (37 mM NaCl, 2.68 mM KCl, 8.1 mM Na_2_HPO_4_, 1.47 mM KH_2_PO_4_, pH 7.4) and immediately dissected under a stereoscope (SMZ800N, Nikon, Japan) using fine insect anatomical needles. In brief, a complete ovary was obtained by cutting the side of the female abdomen and removing the abdominal exoskeleton and other tissues. Subsequently, a typical ovary was photographed, and a hand drawing of the ovary was created by tracing on sulfuric acid paper (Figure 6).

### 4.3. DsRNA Synthesis

Ovaries of *D. citri* were dissected as previously described and immediately frozen in liquid nitrogen in a 1.5 mL Eppendorff tube for total RNA extraction using TransZol Up (TransGen Biotech, Beijing, China). Then, cDNA synthesis was carried out using the PrimescriptTM RT reagent kit with gDNA eraser (perfect real time) (TaKaRa, Tokyo, Japan) following the manufacturer’s instructions: (1) prepare the following mixture in an RNase-free Eppendorff tube, RNase-free ddH_2_O to 16 µL, 4 µL 4 × 2-Step Gdna Erase-Out Mix, total RNA 1 pg-1 µg, 42 °C reaction for 2 min; (2) add 4 µL 5× ToloScript qRT EasyMix directly to the reaction solution in step 1, conduct reverse transcription reaction, 37 °C for 15 min, 85 °C for 5 s. *Vg4* and *VgR* gene interference fragments were designed using E-RNAi Webservice https://www.dkfz.de/signaling/e-rnai3/idseq.php (accessed on 15 February 2023) (E-RNAi team, Thomas Horn and Michael Boutros). Using the nucleotide sequences of the *Vg4* and *VgR* genes, primers were designed for PCR, quantitative real-time PCR (qRT–PCR), and dsRNA synthetic primers in NCBI (National Center for Biotechnology Information, Bethesda, Maryland, USA) (Table 3). *Vg4* (477 bp) cDNA fragments were generated with the primer pair *Vg4* F3/R3 using 2 × FastPfu Premix (TOLOBIO, Shanghai, China). The purified PCR *Vg4* fragments were inserted into the plasmid of the pMD18T vector (DH5α, Thermo Fisher Scientific, Waltham, Massachusetts, USA). The resulting *Vg4* plasmids were used as templates to generate ds*Vg4* with the dsRNA synthesis kit (RNAsyn Biottech Co., Ltd., Jiangsu, Suzhou, China; Appendix A), following the manufacturer’s instructions: the preparation of dsRNA synthesis (DEPC water 51.6 μL, 5× transcription buffer 24 μL, NTP 20.4 μL, transcriptional template 6 μL, RNase enzyme inhibitor 3.6 μL, DTT 2.4 μL, RNA polymerase 12 μL), transcript at 37 °C for 1 h, add 12 μL DNaseI and react at 37 °C for 30 min, remove the template, and purify using NEB’s RNA purification kit (NEB, Ipswich, Massachusetts, USA) to obtain positive and negative *Vg4* RNA; mix positive and negative *Vg4* RNA in equal proportions and anneal to obtain ds*Vg4*, with detailed synthesis methods provided in the Appendix A. Synthesized dsRNA was quantitated by an N80 Touch nanophotometer (Implen, Munich, Bavaria, Germany) at 260 nm, and the integrity was analyzed by agarose gel electrophoresis. The syntheses of ds*VgR* (346 bp) and dsGPF (726 bp) were carried out as described for ds*Vg4* (Table 3).

### 4.4. Delivery and Stability of dsVg4 and dsVgR in M. odorifera Shoots

Six *M. odorifera* shoots were immersed in a plastic container containing ds*Vg4* solution (1 mL) at a concentration of 10 ng/µL. The ds*Vg4* solution in the Eppendorff tubes was maintained at 1 mL by adding water. Individual *M. odorifera* shoot and leaf samples were collected once a day for six days, immediately frozen in liquid nitrogen, and stored at −80 °C until RNA extraction. Total RNA and cDNA from shoots and leaves of *M. odorifera* were sequentially prepared as previously described. The cDNA, used as a template, was employed to clone *Vg4*, and the PCR product was visualized by gel electrophoresis as described previously (repeated at least six times). Clear bands indicated that ds*Vg4* was stable and transferred to shoots and leaves as expected. The stability of ds*VgR* was the same as that of ds*Vg4*. The clear bands of *Vg4* and *VgR* during gel electrophoresis were purified using the EasyPure Quick Gel Extraction Kit (Transgen, Beijing, China) for sequencing (RNAsyn Biottech Co., Ltd., Jiangsu, Suzhou, China) to verify whether the cloned fragment base sequences corresponded to dsRNA. The sequenced data were analyzed by Chromas (Technelysium Pty Ltd., South Brisbane, Queensland, Australia). Ds*GFP*-designed primers were used for independent stability detection; the stability detection method for ds*GFP* was identical to that of ds*Vg4*.

### 4.5. Interference Efficiency of dsVg4 and dsVgR Treatment and Ovarian Morphological Changes in D. citri

Six rearing containers were allocated to each of the ds*Vg4*, ds*VgR*, and ds*GFP* treatments. Fifteen one-day-old females and males were randomly selected, paired, and reared in each container as previously described. Females from one container per treatment were selected every 5 days and dissected for ovary sampling, while *M. odorifera* tender shoots and ds*Vg4* solution were replaced in other containers. The dissected ovaries were first photographed with a microscopic measurement system coupled to a stereoscope (SMZ800N, Nikon, Japan) for further egg measurements (Figure 7) and then sampled for RNA extraction. In this way, ovaries were collected from 5-, 10-, 15-, 20-, 25-, and 30-day-old females that were treated with ds*Vg4*, ds*VgR*, and ds*GFP*. The cDNA from these samples was prepared as previously described and used for qRT–PCR assays to check RNAi efficiency.

qRT-PCR was carried out using 2 × Q3 SYBR qPCR Master Mix (High Rox) (TOLOBIO, Shanghai, China) following the manufacturer’s instructions: (1) prepare the following mixture in an RNase-free Eppendorff tube, 2 × Q3 SYBR qPCR Master Mix 10 μL, primer 1 (10 µM) 0.4 μL, primer 2 (10 µM) 0.4 μL, template cDNA 1 μL, ddH_2_O to 20 μL; (2) place the prepared solution into 96-well plates and perform qRT-PCR on the qRT-PCR machine. The qRT-PCR reaction conditions were as follows: pre-denaturation, cycle 1 at 95 °C for 30 s; circular reaction, cycle 40 at 95 °C for 10 s; melting curves, cycle 1 at 95 °C for 15 s, 60 °C for 60 s, and 95 °C for 15 s. All reactions were performed with the QuantStudioTM Real-Time PCR system (Applied Biosystems, Waltham, MA, USA) using the primers listed in Table 1. The expression analyses of ds*VgR* and ds*GFP* were the same as that of ds*Vg4*. Relative gene expression data from qRT-PCR were analyzed using the 2^−ΔΔCT^ method.

### 4.6. Interference of Vg and VgR on Oviposition in D. citri

Newly emerged, unmated females (n = 30) were randomly selected, fed ds*Vg4* (300 ng), and paired with 7- to 12-day-old unmated males (n = 30). A pair of male and female was placed in each rearing container. The ds*VgR*, ds*GFP*, and blank control (water) treatments were carried out as described for the ds*Vg4* treatment. Each day, *M. odorifera* shoots were replaced when eggs were detected on the shoots. The replaced *M. odorifera* shoots were stored separately after counting the number of eggs. The number of nymphs was also counted after egg hatching. The fecundity, egg hatchability, and deformity rate of eggs and nymphs were assessed after all females had died (the lifetime of *D. citri* females is 30–90 days).

### 4.7. Statistical Analysis

Data analysis was performed using SPSS 25.0 (IBM Corp., Armonk, New York, NY, USA). Data on the length and width of eggs, number of mature eggs, mature egg percentage, number of eggs per female, and egg hatchability were analyzed using one-way analysis of variance (ANOVA) followed by Tukey’s honestly significant difference (HSD) multiple tests. The effects of ds*GFP* and ds*Vg4* on the relative expression of *Vg4* and *VgR* were analyzed using independent samples *t*-tests. The difference was considered statistically significant at the 5% level (*p* < 0.05).

## 5. Conclusions

In summary, interference with *Vg4* and *VgR* gene expression can prevent the accumulation of *Vitellogenin* in the ovary of *D. citri*, leading to an inability to form mature eggs normally. Consequently, most eggs remain immature and egg formation is suppressed, resulting in the production of abnormal eggs and nymphs. The expression of the *Vg4* and *VgR* genes was significantly affected by ds*Vg4* and ds*VgR*. Ds*VgR* was more effective than ds*Vg4* at suppressing egg formation in *D. citri*, suggesting that ds*VgR* is more suitable than ds*Vg4* for use in the integrated management of insect populations.

## Figures and Tables

**Figure 1 ijms-24-09497-f001:**
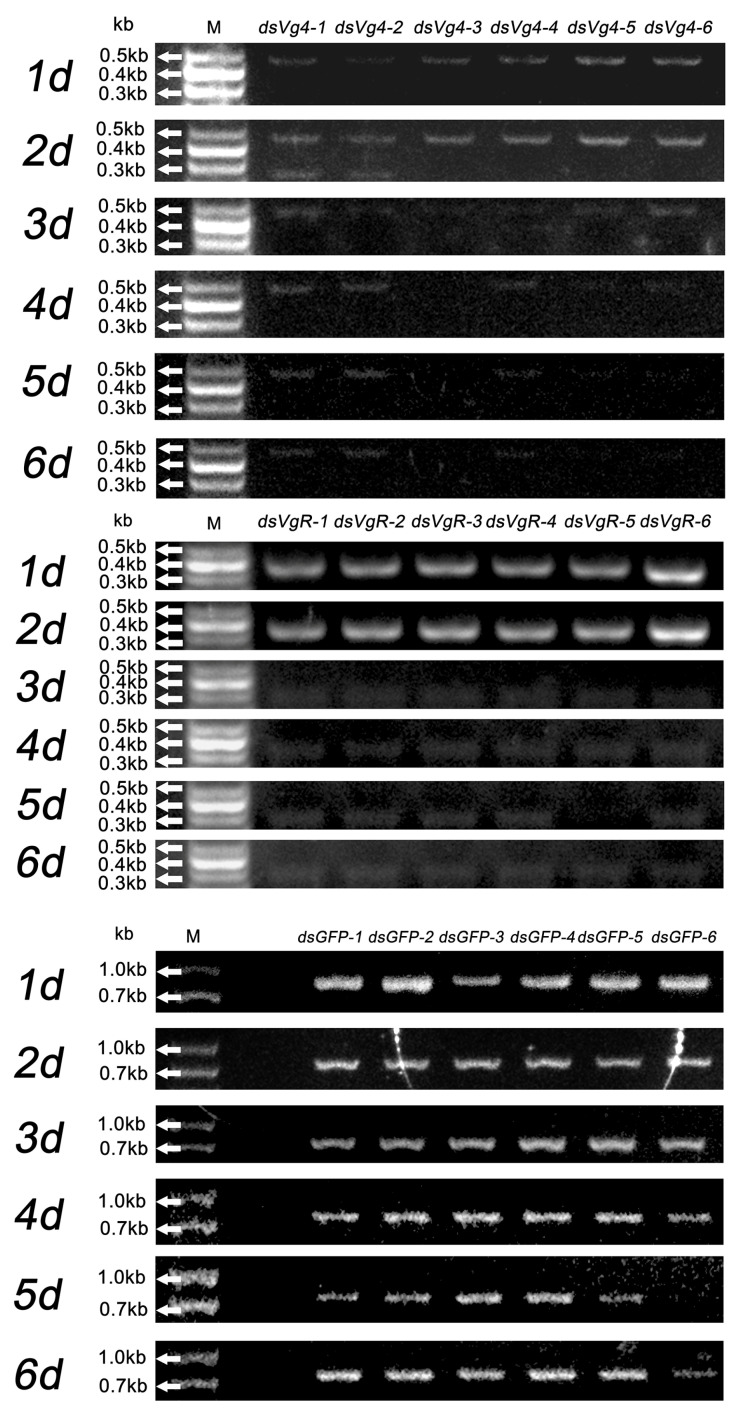
Ds*Vg4*, ds*VgR*, and ds*GFP* were absorbed by tender *M. odorifera* shoots for six days, and gel electrophoresis was used to detect the persistence of ds*Vg4*, ds*VgR*, and ds*GFP* after each day. The gel electrophoresis bands of ds*Vg4* were relatively clear at 1–6 d; the gel electrophoresis bands of ds*VgR* were relatively clear at 1–2 d, but blurry at 3–6 d; the gel electrophoresis bands of ds*GFP* were relatively clear at 1–6 d. The ds*Vg4*-1, 2, 3, 4, 5, 6/ds*VgR*-1, 2, 3, 4, 5, 6/ds*GFP*-1, 2, 3, 4, 5, 6 represent the six replicates for each treatment.

**Figure 2 ijms-24-09497-f002:**
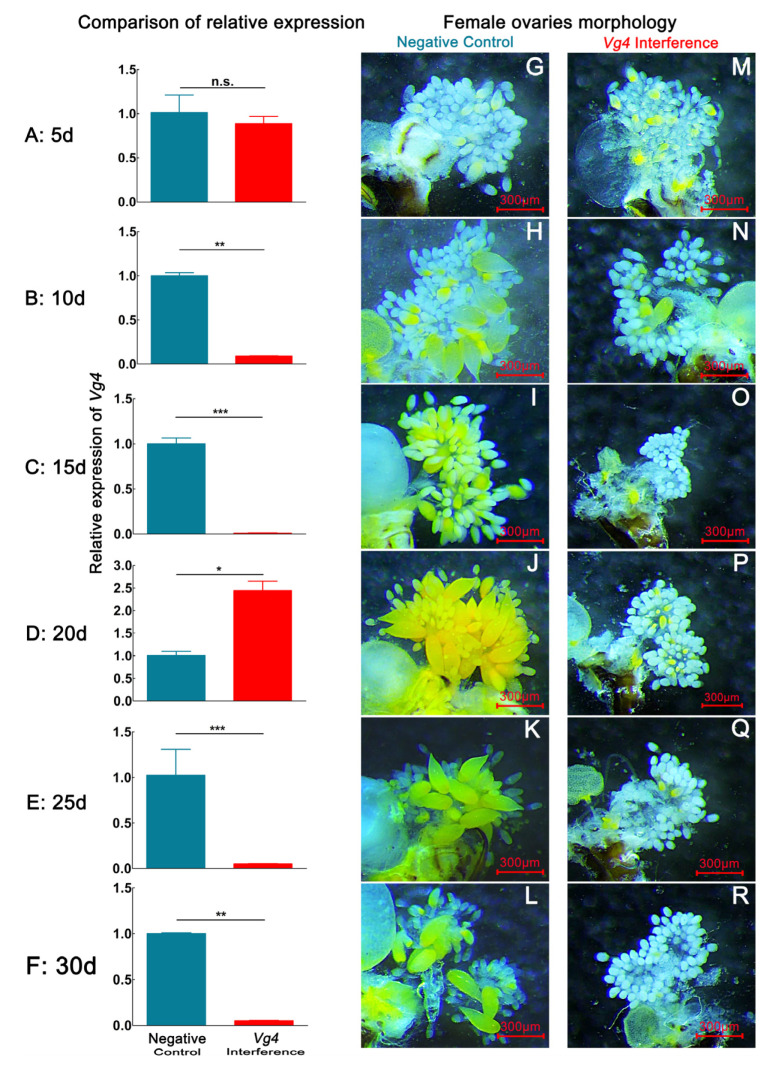
Interference effect of ds*GFP* and ds*Vg4* via continuous ds*Vg4* feeding over 30 d on the *Vg4* expression and ovary developmental morphological characteristics in *D. citri*. (**A**,**G**,**M**), (**B**,**H**,**N**), (**C**,**I**,**O**), (**D**,**J**,**P**), (**E**,**K**,**Q**), (**F**,**L**,**R**) represent the relative expression level of *Vg4* and ovarian morphology of *Vg4* interference-treated and negative-control groups on the 5th, 10th, 20th, 25th, and 30th days, respectively. Data are shown as the mean ± SD. Values of *P* were based on the independent samples *t* test: *** *p* < 0.001; ** *p* < 0.01; * *p* < 0.05; ns, *p* ≥ 0.05. The eggs have a yellow color. The blue color in the bar chart represents the negative control group, and the red color represents the *Vg4* interference group. (**G**–**L**) correspond to the ovaries of negative control group females. (**M**–**R**) correspond to ovaries of *Vg4* interference group female.

**Figure 3 ijms-24-09497-f003:**
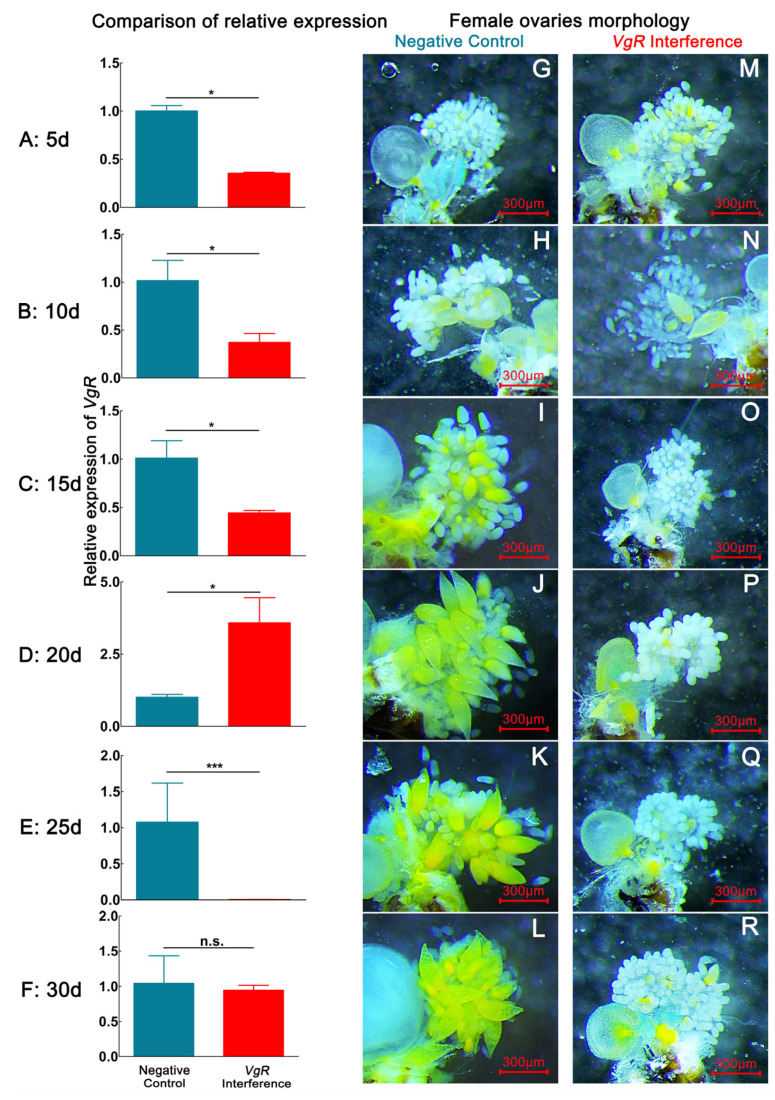
The interference effect of ds*GFP* and ds*VgR* on *VgR* expression and ovary developmental morphological characteristics in *D. citri* via continuous ds*VgR* feeding within 30 d. (**A**,**G**,**M**), (**B**,**H**,**N**), (**C**,**I**,**O**), (**D**,**J**,**P**), (**E**,**K**,**Q**), and (**F**,**L**,**R**) represent the relative expression level of *VgR* and ovarian morphology of *VgR* interference-treated and negative-control groups on the 5th, 10th, 20th, 25th, and 30th days, respectively. Data are shown as the mean ± SD. *P* values are from the independent samples *t* test: *** *p* < 0.001; * *p* < 0.05; ns, *p* ≥ 0.05. The eggs have a yellow color. The blue color in the bar chart represents the negative control group, the red color represents the *VgR* interference group. (**G**–**L**) correspond to the ovaries of negative control group female. (**M**–**R**) correspond to the ovaries of *VgR* interference group female.

**Figure 4 ijms-24-09497-f004:**
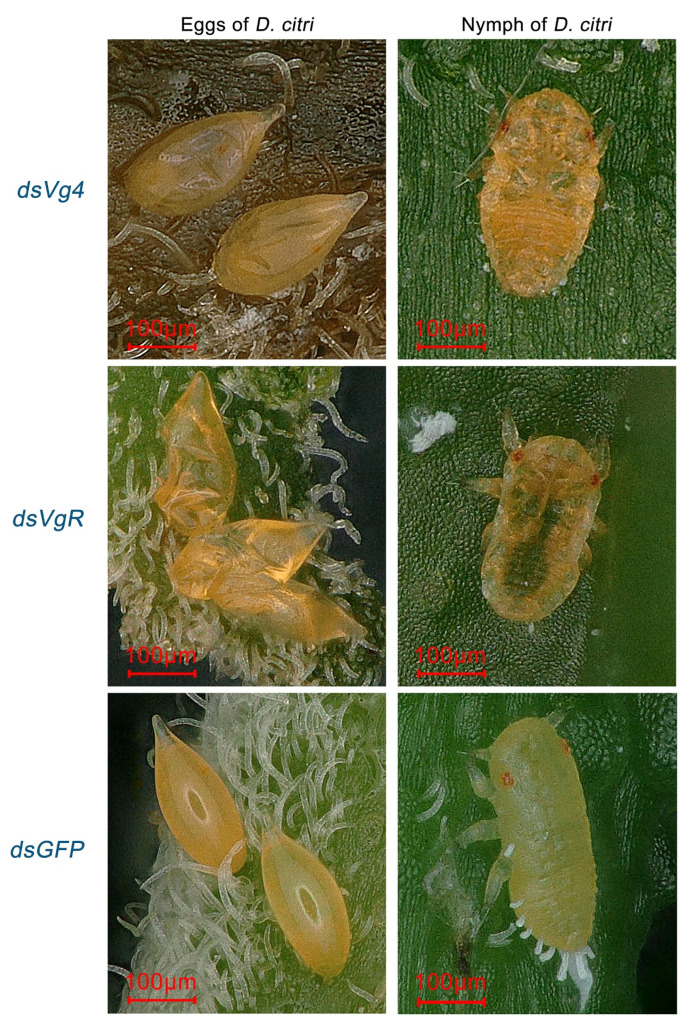
Typical abnormal shriveled eggs and wrinkled nymphs of the *Vg4* and *VgR* interference groups in *D. citri*. Eggs in ds*GFP* have a water droplet-like shape with a smooth, yellow surface. Eggs in the *Vg4* and *VgR* interference groups appeared shrunken and brown compared to normal eggs, with an abnormal surface. Nymphs in ds*GFP* have a yellow body with a smooth back, while abnormal nymphs in the *Vg4* and *VgR* interference groups showed wrinkled backs with a brown-yellow body color, and some nymphs in the *Vg4* interference group had a semi-transparent body. Abnormal eggs did not hatch, and abnormalities were observed in the nymphs at the 1–2 instar stage.

**Figure 5 ijms-24-09497-f005:**
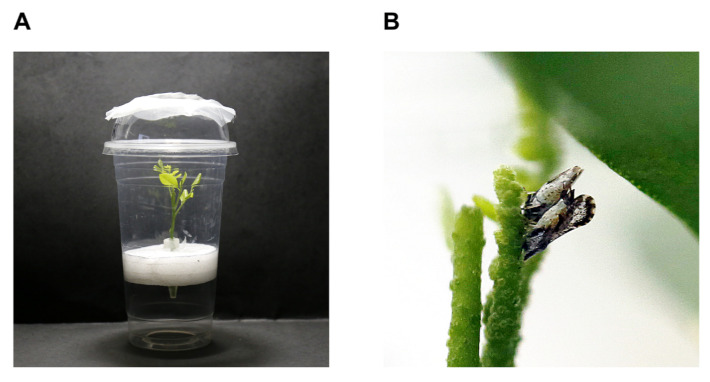
Insect rearing device. (**A**) Plastic container with *M. odorifera*; and (**B**) Mating of male and female adults on *M. odorifera*.

**Figure 6 ijms-24-09497-f006:**
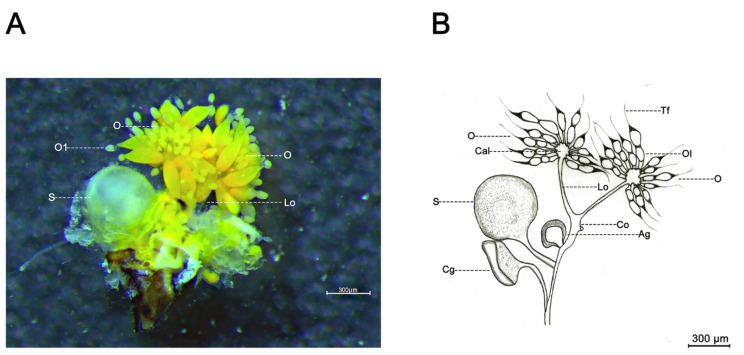
Ovarian morphology of *D. citri* female. (**A**) Ovarian morphology; and (**B**) Hand drawing of ovarian morphology. Ag, accessory glands; Cal, calyx; Cg, colleterial gland; Co, common oviduct; Lo, lateral oviduct; O, ovary; Ol, ovariole; S, spermatheca; Tf, terminal filament.

**Figure 7 ijms-24-09497-f007:**
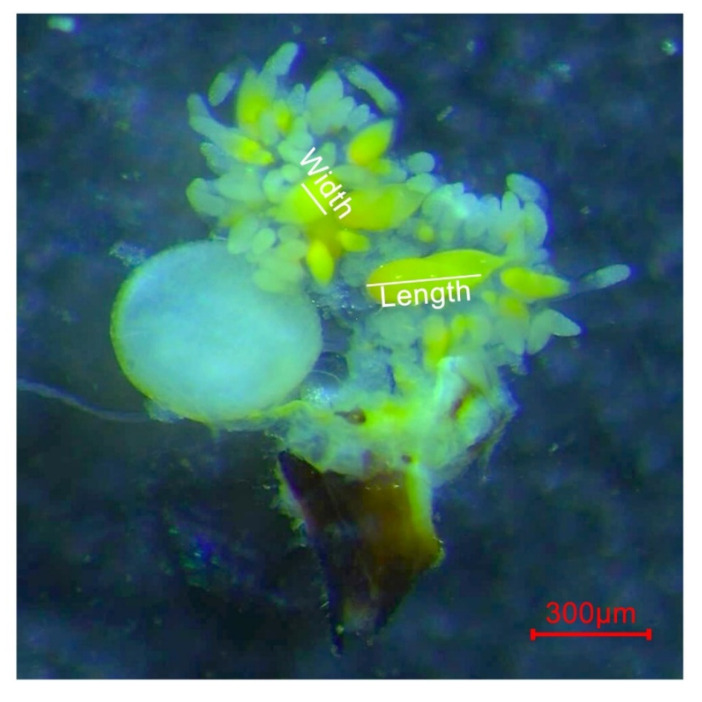
Measurement method of *D. citri* eggs. The length and width of the eggs are marked with white lines.

**Table 1 ijms-24-09497-t001:** Comparison of egg size and number of mature eggs of *D. citri* in different treatments. The egg length and width, number of eggs, and proportion of mature eggs were compared between 1 and 30 d. Data from the same day were compared vertically.

Day (d)	Treatments	Egg Length (µm)	Egg Width (µm)	Number of Mature Eggs	Mature Eggs Percentage (%)
5 d	Ds*Vg4*	0.00 ± 0.00 a	0.00 ± 0.00 a	0.00 ± 0.00 a	0.00 ± 0.00 a
	Ds*VgR*	0.00 ± 0.00 a	0.00 ± 0.00 a	0.00 ± 0.00 a	0.00 ± 0.00 a
	Ds*GFP*	0.00 ± 0.00 a	0.00 ± 0.00 a	0.00 ± 0.00 a	0.00 ± 0.00 a
10 d	Ds*Vg4*	136.30 ± 15.55 a	62.58 ± 6.30 a	1.03 ± 0.37 a	8.33 ± 1.67 b
	Ds*VgR*	123.32 ± 30.03 a	61.62 ± 6.59 a	0.75 ± 0.25 a	7.75 ± 0.5 b
	Ds*GFP*	142.67 ± 45.99 a	66.51 ± 21.81 a	1.29 ± 0.46 a	13.19 ± 0.14 a
15 d	Ds*Vg4*	47.99 ± 3.99 b	28.02 ± 3.54 b	0.00 ± 0.00 b	0.00 ± 0.00 b
	Ds*VgR*	0.00 ± 0.00 c	0.00 ± 0.00 c	0.00 ± 0.00 b	0.00 ± 0.00 b
	Ds*GFP*	249.16 ± 50.81 a	108.29 ± 11.35 a	5.33 ± 1.33 a	16.08 ± 5.80 a
20 d	Ds*Vg4*	71.26 ± 13.52 b	41.63 ± 4.97 b	0.00 ± 0.00 b	0.00 ± 0.00 b
	Ds*VgR*	79.78 ± 9.61 b	46.06 ± 8.52 b	0.00 ± 0.00 b	0.00 ± 0.00 b
	Ds*GFP*	285.83 ± 37.97 a	109.52 ± 8.12 a	13.25 ± 0.75 a	52.91 ± 19.29 a
25 d	Ds*Vg4*	75.17 ± 9.90 b	43.79 ± 6.99 b	0.00 ± 0.00 b	0.00 ± 0.00 b
	Ds*VgR*	0.00 ± 0.00 c	0.00 ± 0.00 c	0.00 ± 0.00 b	0.00 ± 0.00 b
	Ds*GFP*	293.71 ± 9.74 a	124.24 ± 18.77 a	11.38 ± 0.88 a	23.36 ± 4.47 a
30 d	Ds*Vg4*	0.00 ± 0.00 c	0.00 ± 0.00 c	0.00 ± 0.00 b	0.00 ± 0.00 b
	Ds*VgR*	70.64 ± 8.87 b	43.96 ± 0.24 b	0.00 ± 0.00 b	0.00 ± 0.00 b
	Ds*GFP*	286.91 ± 37.67 a	115.77 ± 7.93 a	9.33 ± 3.00 a	18.91 ± 6.42 a

Means ± SDs followed by different lowercase letters in the columns on the same days are significantly different (Tukey, *p* < 0.05).

**Table 2 ijms-24-09497-t002:** Comparison of *D. citri* oviposition and nymph formation in interference group and control group. The items compared were the total number of eggs deposited, number of eggs deposited per female, total number of nymphs, egg hatchability, egg deformity, and nymph deformity. Units of comparison items are provided in parentheses.

Treatment	Fecundity	Number of Eggs/Female	Total Number of Nymphs	Egg Hatchability (%)	Egg Deformity (%)	Nymph Deformity (%)
Ds*Vg4*	1379	45.97 ± 7.31 b	1190	84.41 ± 6.50 b	4.42	4.12
Ds*VgR*	638	20.80 ± 9.37 c	594	81.11 ± 7.30 b	6.80	1.35
Ds*GFP*	2804	93.47 ± 10.20 a	2570	94. 89 ± 2.76 a	0	0
Blank control	2457	84.97 ± 12.28 a	2380	96.67 ± 1.94 a	0	0

Means ± SDs followed by different lowercase letters in columns are significantly different (Tukey, *p* < 0.05).

**Table 3 ijms-24-09497-t003:** Oligonucleotide primer pairs used in this study.

Gene	Primer Name	Sequences of Primers (5′→3′)	Application
*Vg4*	Vg4 F1	CGCCAGGGTTTTCCCAGTCACGAC	dsRNA synthesis positive-F
	Vg4 R1	GGGCAACCATTGAAAGACGTTGGAAG	
	Vg4 F2	GGGATGGCCATGAAACAATGGATTG	dsRNA synthesis negative-R
	Vg4 R2	GTATGTTGTGTGGAATTGTGAGCGG	
	Vg4 F3	ATGGCCATGAAACAATGGATTGA	PCR
	Vg4 R3	GGCAACCATTGAAAGACGTTGGA	
	Vg4 F4	GCCAGATACCCAACCCGTGAATAC	qRT-PCR
	Vg4 R4	AGGATAGCAGAGGTGTTGAGGTGAG	
*VgR*	VgR F1	CGCCAGGGTTTTCCCAGTCACGAC	dsRNA synthesis positive-F
	VgR R1	GGGAAAACTCGGAACATGGCAACAC	
	VgR F2	GGGATGGCAATGATGACTGTGGTG	dsRNA synthesis negative-R
	VgR R2	GTATGTTGTGTGGAATTGTGAGCGG	
	VgR F3	GATGGCAATGATGACTGTGGTGA	PCR
	VgR R3	AAAACTCGGAACATGGCAACACA	
	VgR F4	ACCTGCCAATGCCAAGTATGAGATG	qRT-PCR
	VgR R4	TTGATGGTCACATAGCCAGGAGTTG	
*GFP*	GFPT7 F1	TTAATTGGGCCACCTATAGGGATGGCTAGCAAAGGAGAAGAACTCTTCACTGGAGTTG	dsRNA synthesis positive-F
	GFP R1	GGGTCAGTTGTACAGTTCATCCATGCCATGTG	
	GFP F2	GGGATGGCTAGCAAAGGAGAAGAACTCTTCAC	dsRNA synthesis negative-R
	GFPT7 R2	TTAATTGGGCCACCTATAGGGTCAGTTGTACAGTTCATCCATGCCATGTGTAATCC	
	GFP F3	GGGATGGCTAGCAAAGGAGAAGAACTCTTCAC	PCR
	GFP F3	GGGTCAGTTGTACAGTTCATCCATGCCATGTG	
*Actin*	Actin F	TGTGACGAAGAAGTTGCTGC	qRT-PCR
	Actin R	TGGGGTATTTCAGGGTCAGG	

The T7 RNA polymerase promoter is underlined.

## Data Availability

Not applicable.

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
