# Peer review of "IPS (In-Plant System) Delivery of Double-Stranded Vitellogenin and Vitellogenin receptor via Hydroponics for Pest Control in Diaphorina citri Kuwayama (Hemiptera: Psyllidae)"

_ijms, 2023, doi:10.3390/ijms24119497_

Round 1

Reviewer 1 Report

Authors describe the use of dsRNA to decrease the expression of Diaphorina citri genes that interfere with egg formation and development.

Here follow some suggestions to improve the manuscript:

Line 94 – VgR gene id was already described in line 73.

Line 102 – Why use the word hydroponics? Could it be absorption or similar? Hydroponics refers to a nutritive solution dissolved in water to feed plants.

Figure S1 – It would be better to show the alignment of the amplified sequences of Vg4 and VgR from the plant with the sequence of the template gene. The same for sequences of Vg4 and VgR in pages 4 and 5 of “Supplementary Materials”.

Figure 1 – The figure needs improvements. Values 1d to 6d seem to refer to the number of days M odorifera was exposed to dsRNA. What about dsVg4-1, dsVg4-2, …etc. Are these technical replicates? Please complete the legend of figure 1.

Figure 2 and Figure 3 – it is not clear which day of the assay each group of images corresponds to. I propose to replace the letters A to F with days 5, 10, up to 30. In the figure caption, also write about the colour of the eggs. Which group of images corresponds to the control trial?

Line 169 – Interference in Vg4 and VgR gene expression was more pronounced at 15 days after treatment. However, authors wrote that the interference was after 1-30 days?

Figure 4 – The legend of the figure needs improvements, figure caption describes only the images of the nymphs exposed to dsVg4 and dsVgR, and not to dsGFP.

Line 252 – please complete the sentence:…affected by dsVg4 abd dsVgR in D. citri.

Line 264 – why are viroids referred to here?

Line 385 – Table 3 is inserted in the text, so should not be referred to be in the supplementary materials.

Line 395 – Please explain what you mean with this text: cDNA was used to run clone Vg4.

Line 399 – Please explain what you mean with this text: fragments cloned on the first day were recycled. All this paragraph should be improved.

Line 413 – In the case of females that were sampled after 30 days of exposure to Murraya that adsorbed dsRNA: a) how long were these plants exposed to dsRNA? b) and at what intervals did you replace the plants where the psyllids feed?

Methods related to dsRNA synthesis and cDNA synthesis should be described in detail in the manuscript.

The Conclusions text should be improved and describe the main changes that occurred in egg formation after females fed on plants containing dsRNA.

Figures referred in the text as supplementary images S1 to S7 do not have a caption. The figures found in the text "Supplementary Materials" and which are a duplicate of figures S1 to S7 all have an incomplete caption.

Author Response

Response to Reviewer 1 Comments

Thank you for your valuable comments on our article. According to your suggestions, we have corrected several points addressed by reviewers in our previous manuscript and supplemented several data. Therefore, extensive revisions were made and we believed that the quality of the manuscript has been greatly improved. The detailed point-by-point responses were listed below.

Authors describe the use of dsRNA to decrease the expression of Diaphorina citri genes that interfere with egg formation and development.

Here follow some suggestions to improve the manuscript:

Line 94 – VgR gene id was already described in line 73.

My Response: The VgR gene id in Line 93 has been removed (Line 92).

Line 102 – Why use the word hydroponics? Could it be absorption or similar? Hydroponics refers to a nutritive solution dissolved in water to feed plants.

My Response: Hydroponics has been replaced with IPS delivery (Line 100).

Figure S1 – It would be better to show the alignment of the amplified sequences of Vg4 and VgR from the plant with the sequence of the template gene. The same for sequences of Vg4 and VgR in pages 4 and 5 of “Supplementary Materials”.

My Response: The alignment of the amplified sequences of Vg4 and VgR from the plant with the sequence of the template gene has been added in figure S1 and the corresponding “Supplementary Materials”.

Figure 1 – The figure needs improvements. Values 1d to 6d seem to refer to the number of days M. odorifera was exposed to dsRNA. What about dsVg4-1, dsVg4-2, …etc. Are these technical replicates? Please complete the legend of figure 1.

My Response: The values of 1d to 6d did refer to the number of days M. odorifera was exposed to dsRNA. We clearly indicated it in the figure title. And more details have been added to the legend of Figure 1 for explanation. The added sentence: The dsVg4-1, 2, 3, 4, 5, 6/dsVgR-1, 2, 3, 4, 5, 6/dsGFP-1, 2, 3, 4, 5, 6 represent the six replicates for each treatment (Line 127).

Figure 2 and Figure 3 – it is not clear which day of the assay each group of images corresponds to. I propose to replace the letters A to F with days 5, 10, up to 30. In the figure caption, also write about the colour of the eggs. Which group of images corresponds to the control trial?

My Response: Day indicators have been added at the corresponding positions in Figs 2 and 3 (Line 143; Line 161). The corresponding sentence has been added to explain the color of eggs, the added sentence: The eggs have a yellow colour (Line 149; Line 167). Legends in figure 2 and 3 have been updated with sentences explaining the experimental treatments corresponding to the images, the added sentences: The blue colour in the bar chart represents the negative control group, the red colour represents the dsVg4 interference group. G, H, I, J, K, L correspond to ovaries of negative control group female. M, N, O, P, Q, R correspond to ovaries of dsVg4 interference group female; The blue color in the bar chart represents the negative control group, the red colour represents the dsVgR interference group. G, H, I, J, K, L correspond to ovaries of negative control group female. M, N, O, P, Q, R correspond to ovaries of dsVgR interference group female (Line 150-152; Line 168-170).

Line 169 – Interference in Vg4 and VgR gene expression was more pronounced at 15 days after treatment. However, authors wrote that the interference was after 1-30 days?

My Response: The phrasing of the sentence has been modified, and the revised version now reads as follows: Vg4 and VgR gene expression in D. citri females was interfered with, resulting in a relatively small number of mature ovarian eggs at 15-30 d stage (Figures 2-3, Table 1) (Line 177-178).

Figure 4 – The legend of the figure needs improvements, figure caption describes only the images of the nymphs exposed to dsVg4 and dsVgR, and not to dsGFP.

My Response: Additional information has been added to the legend of Figure 4 to explain the differences in phenotypic features between abnormal eggs and nymphs in the interference group and dsGFP eggs and nymphs (Line 204-209). The added sentence: Eggs in dsGFP have a water droplet-like shape with a smooth, yellow surface. Eggs in dsVg4 and dsVgR interference groups appeared shrunken and brown compared to normal eggs, with an abnormal surface. Nymphs in dsGFP have a yellow body with a smooth back, while abnormal nymphs in the dsVg4 and dsVgR interference groups showed wrinkled backs with a brown-yellow body colour, and some nymphs in the dsVg4 group had a semi-transparent body. Abnormal eggs did not hatch, and abnormalities were observed in nymphs at the 1-2 instar stage.

Line 252 – please complete the sentence:…affected by dsVg4 abd dsVgR in D. citri.

My Response: The content of sentence has been completed, the revised sentence: Gene expression data showed that Vg4 and VgR expression levels were significantly affected by dsVg4 and dsVgR in D. citri (Line 265-266).

Line 264 – why are viroids referred to here?

My Response: Viroids have been removed, the revised sentence: Whether the long-term feeding of dsRNA leads to the long-term activation of immune response remains unclear, and research on the resistance of pests to dsRNA pesticides is not comprehensive (Line 278).

Line 385 – Table 3 is inserted in the text, so should not be referred to be in the supplementary materials.

My Response: Accept account the comments of reviewer, Table 3 is not mentioned in the supplementary materials.

Line 395 – Please explain what you mean with this text: cDNA was used to run clone Vg4.

My Response: The word ‘run’ has been deleted from the sentence, the modified sentence: The obtained cDNA was used to clone Vg4, and the PCR product was visualized by gel electrophoresis as described above (repeat at least 6 times) (Line 418-420).

Line 399 – Please explain what you mean with this text: fragments cloned on the first day were recycled. All this paragraph should be improved.

My Response: The sentence has been modified, the modified sentence: The clear bands of Vg4 and VgR during gel electrophoresis were recycled through EasyPure Quick Gel Extraction Kit (Transgen, China) for sequencing (RNAsyn Biottech Co., Ltd, China) to check whether the cloned fragments base sequence corresponding to dsRNA. The sequenced data were analysed by Chromas (Technelysium Pty Ltd, Australia) (Line 421-425).

Line 413 – In the case of females that were sampled after 30 days of exposure to Murraya that adsorbed dsRNA: a) how long were these plants exposed to dsRNA? b) and at what intervals did you replace the plants where the psyllids feed?

My Response: The corresponding content has been added in the sentence, the modified sentence: Females from one container per treatment were selected every 5 d and dissected for ovary sampling, and replace the M. odorifera tender shoots and dsVg4 solution in other containers (Line 433-434).

Methods related to dsRNA synthesis and cDNA synthesis should be described in detail in the manuscript.

My Response: The synthesis methods of cDNA and dsRNA have been added to the corresponding positions in the article (Line 385-389; Line 398-406).

The Conclusions text should be improved and describe the main changes that occurred in egg formation after females fed on plants containing dsRNA.

My Response: The corresponding content has been added in the conclusion section, the added content: In summary, interference with Vg4 and VgR gene expression can prevent the accumulation of vitellogenin in the ovary of D. citri, which causes ovaries to be unable to form mature eggs normally, most of the eggs were immature and the formation of eggs was suppressed, and leads to the production of abnormal eggs and nymphs (Line 468-470).

Figures referred in the text as supplementary images S1 to S7 do not have a caption. The figures found in the text "Supplementary Materials" and which are a duplicate of figures S1 to S7 all have an incomplete caption.

My Response: Caption of images S1-S7 is on line 474-499, the content of legend has been improved, images S1 to S7 has been explained in detail in the supplementary materials word file.

Reviewer 2 Report

The authors used RNAi technology through hydroponics to suppress the formation of D. citri eggs. They tested over a 30 days time-period whether dsVg4 and dsVgR could effectively reduce the expression level of the Vg4 and VgR genes of D. citri. They selected Vg4 protein gene because it had the most complex amino acid structure in the Vg gene family. Overall, dsVgR was more effective than dsVg4 in suppressing egg formation. The study is valuable and add to our knowledge in the field of biological control of D. citri.

For authors:

1.     Grammar corrections are required, for example, lines 315, 391, ..etc.

2.     In Fig. 1, use kb instead of bp to refer to the size of the marker band.

3.     I recommend adding the day (i.e. 5-30 days) next to the y-axis in Figs 2 and 3 to make it easier to track changes in gene expression or ovary over time.

4.     In Fig. 2 B, C, E, and F, the scale in the Y-axis needs to be adjusted.

5.     In Fig. 3 E, the scale in the Y-axis needs to be adjusted.

6.     Rewrite the legend for Figure 1 and add more details. It is necessary for understanding the figure without having to return to the main text.

7.     Please add more details to the legend of Fig. 4 to refer to control versus treatment changes.

Author Response

Response to Reviewer 2 Comments

Thank you for your valuable comments on our article. According to your suggestions, we have corrected several points addressed by reviewers in our previous manuscript and supplemented several data. Therefore, extensive revisions were made and we believed that the quality of the manuscript has been greatly improved. The detailed point-by-point responses were listed below.

The authors used RNAi technology through hydroponics to suppress the formation of D. citri eggs. They tested over a 30 days time-period whether dsVg4 and dsVgR could effectively reduce the expression level of the Vg4 and VgR genes of D. citri. They selected Vg4 protein gene because it had the most complex amino acid structure in the Vg gene family. Overall, dsVgR was more effective than dsVg4 in suppressing egg formation. The study is valuable and add to our knowledge in the field of biological control of D. citri.

For authors:

  1. Grammar corrections are required, for example, lines 315, 391, ..etc.

My Response: Sentences with grammar issues have been modified (Line 330; Line 417).

  1. In Fig. 1, use kb instead of bp to refer to the size of the marker band.

My Response: The bp in Fig. 1 has been modified to kb (Line 122).

  1. I recommend adding the day (i.e. 5-30 days) next to the y-axis in Figs 2 and 3 to make it easier to track changes in gene expression or ovary over time.

My Response: Day indicators have been added at the corresponding positions in Figs 2 and 3 (Line 143; Line 161).

  1. In Fig. 2 B, C, E, and F, the scale in the Y-axis needs to be adjusted.

My Response: The Y-axis in Fig. 2 B, C, E, and F have been adjusted (Line 143).

  1. In Fig. 3 E, the scale in the Y-axis needs to be adjusted.

My Response: The Y-axis in Fig. 3 E have been adjusted (Line 161).

  1. Rewrite the legend for Figure 1 and add more details. It is necessary for understanding the figure without having to return to the main text.

My Response: The legend in Figure 1 has been updated with corresponding information (Line 123-128). The added sentence: The gel electrophoresis bands of dsVg4 were relatively clear at 1-6 d; the gel electrophoresis bands of dsVgR were relatively clear at 1-2 d, but blurry at 3-6 d; the gel electrophoresis bands of dsGFP were relatively clear at 1-6 d. The dsVg4-1, 2, 3, 4, 5, 6/dsVgR-1, 2, 3, 4, 5, 6/dsGFP-1, 2, 3, 4, 5, 6 represent the six replicates for each treatment.

  1. Please add more details to the legend of Fig. 4 to refer to control versus treatment changes.

My Response: Additional information has been added to the legend of Figure 4 to explain the differences in phenotypic features between abnormal eggs and nymphs in the interference group and normal eggs and nymphs (Line 203-209). The added sentence: Typical abnormal shrivelled eggs and wrinkled nymphs of the dsVg4 and dsVgR interference groups in D. citri. Eggs in dsGFP have a water droplet-like shape with a smooth, yellow surface. Eggs in dsVg4 and dsVgR interference groups appeared shrunken and brown compared to normal eggs, with an abnormal surface. Nymphs in dsGFP have a yellow body with a smooth back, while abnormal nymphs in the dsVg4 and dsVgR interference groups showed wrinkled backs with a brown-yellow body colour, and some nymphs in the dsVg4 group had a semi-transparent body. Abnormal eggs did not hatch, and abnormalities were observed in nymphs at the 1-2 instar stage.

Round 2

Reviewer 1 Report

Amendments have been made to the text, but the text is not clear and still needs improvements. Authors should revise carefully the text and ask for the help of someone with a high command in English to revise the text too.

Suggestions to improve the manuscript:

Figures 1 and 2 are repeated and a red bar is over some images.

Line 82-83 – The sentence is confuse, please revise it.

Line 88 – Vg4 was previously described in line 70.

Line 116 – The control group is referred with the word “contrast” in Fig. S2, S3, S4. “Contrast” sample and control group are the same? If not, please explain what is the “contrast” sample. Did the authors amplify dsGFP group and the blank control group with the Vg4 and VgR PCR primers to show that the PCR was specific for amplification of the Vg4 and VgR genes? If so, please clarify in the text.

Line 131 – Authors wrote that “the expression of the Vg4 gene was abnormally increased at 20 d, and the relative expression increased by 143.65%”. These results are confirmed in Figure 2 where the expression of the control group (dsGFP) was assigned the value 1 to determine the relative expression of Vg4 in ovaries of D. citri females that feed on plants inoculated with dsVg4 (this information is not clear in the text). In this case, the figure legend must be modified because the relative expression was determined for Vg4 and not for dsVg4. The same modification should be done in Figure 3. Please be more specific in your descriptions in the text. In the Materials and Methods section, it should be described how was performed the relative quantification of Vg4 and VgR by real-time PCR.

Line 230 - the word “hydroponics” appears again in discussion section.

Line 250 – The authors wrote that “The reverse transcription reaction can be carried out when the dsRNA 250 becomes single-stranded”. Please explain the context of this sentence as eukaryotic cells do not have reverse transcriptase.

Line 385 – Authors did not correct the following information: Table 3 is inserted in the text, therefore it is not included in the supplementary materials.

There are several expressions which are not common, as the description that “The obtained cDNA was used to clone Vg4”. cDNA is used to amplify a sequence by PCR that is cloned into a plasmid. About the sentence “The clear bands of Vg4 and VgR during gel electrophoresis were recycled”, the usual term is that the bands are purified.

Supplementary material

1.4 RT reaction: please write cDNA and not CDNA.

Page 4 – Please explain the gel electrophoresis results on page 4. The legend of the figure indicates the detection of dsRNA, but in the figure is seen VgR and Vg4.

Author Response

Response to Reviewer 1 Comments

Dear Reviewer,

Thank you for your valuable comments on our article. According to your suggestions, we have corrected again several points you addressed based on our previous manuscript. With the help of USA peer scientist in the same field, the revised part of manuscript has been greatly improved. The detailed point-by-point responses were listed below.

Amendments have been made to the text, but the text is not clear and still needs improvements. Authors should revise carefully the text and ask for the help of someone with a high command in English to revise the text too.

My Response: The article has been carefully revised, and experts with higher English proficiency have revised article again who is a peer scientist in the USA.

Suggestions to improve the manuscript:

Figures 1 and 2 are repeated and a red bar is over some images.

My Response: After careful checking, we confirm that Figure 1 and 2 are not repeated. Secondly, the red bar has been removed by accepting the modifications presented herein (Line 124; Line 145; Line 163).

Line 82-83 – The sentence is confuse, please revise it.

My Response: The sentence with incorrect expression has been modified. The revised sentence: Instead, direct oral feeding of dsRNA on D. citri was mainly applied to verify the effect or function of relative genes for a period of time [20-21] (Line 83-85).

Line 88 – Vg4 was previously described in line 70.

My Response: The duplicate content in the sentence has been deleted, the modified sentence: We further discovered that the Vg4 protein, corresponding to the Vg4 gene, had the most complex amino acid structure in the Vg gene family and was highly expressed in the female abdomen (Line 89-91).

Line 116 – The control group is referred with the word “contrast” in Fig. S2, S3, S4. “Contrast” sample and control group are the same? If not, please explain what is the “contrast” sample. Did the authors amplify dsGFP group and the blank control group with the Vg4 and VgR PCR primers to show that the PCR was specific for amplification of the Vg4 and VgR genes? If so, please clarify in the text.

My Response: The dsGFP group and blank control group parts in the legends of Fig. S2/S3/S4 have been uniformly modified. DsGFP group and the blank control group with the Vg4 and VgR PCR primers to show that the PCR was specific for amplification of the Vg4 and VgR genes, which has been supplemented in the article, the sentence after content supplementation: Gel electrophoresis was also performed on the dsGFP group and blank control group using Vg4 and VgR PCR primers to demonstrate the reliability of the PCR results. No clear bands were observed in the gel electrophoresis when Vg4 and VgR PCR primers were used in the PCR process for the dsGFP group and blank control group, proving the specificity of Vg4 and VgR PCR primers (Figure S2) (Line 117-122; Line 489-497).

Line 131 – Authors wrote that “the expression of the Vg4 gene was abnormally increased at 20 d, and the relative expression increased by 143.65%”. These results are confirmed in Figure 2 where the expression of the control group (dsGFP) was assigned the value 1 to determine the relative expression of Vg4 in ovaries of D. citri females that feed on plants inoculated with dsVg4 (this information is not clear in the text). In this case, the figure legend must be modified because the relative expression was determined for Vg4 and not for dsVg4. The same modification should be done in Figure 3. Please be more specific in your descriptions in the text. In the Materials and Methods section, it should be described how was performed the relative quantification of Vg4 and VgR by real-time PCR.

My Response: The figure 2-3 and legends have been modified as required, and the specific qRT‒PCR method has been included in the materials and methods section (Line 139; Line 149; Line 152-154; Line 162; Line167-172; Line 444-450).

Line 230 - the word “hydroponics” appears again in discussion section.

My Response: The hydroponics in the sentence has been replaced with IPS method (Line 231).

Line 250 – The authors wrote that “The reverse transcription reaction can be carried out when the dsRNA 250 becomes single-stranded”. Please explain the context of this sentence as eukaryotic cells do not have reverse transcriptase.

My Response: Reverse transcription reaction is achieved by extracting RNA from M. odorifera shoots and using cDNA reagents, and which does not mean that M. odorifera cells undergo reverse transcription reaction. The sentence expression has been modified, the modified sentence: The cDNA and PCR reagents can be used to detect the presence of dsRNA when the dsRNA in plants becomes single-stranded (Line 251-252).

Line 385 – Authors did not correct the following information: Table 3 is inserted in the text, therefore it is not included in the supplementary materials.

My Response: The supplementary materials in the sentence have been deleted (Line 411).

There are several expressions which are not common, as the description that “The obtained cDNA was used to clone Vg4”. cDNA is used to amplify a sequence by PCR that is cloned into a plasmid. About the sentence “The clear bands of Vg4 and VgR during gel electrophoresis were recycled”, the usual term is that the bands are purified.

My Response: The incorrect expression in the sentence has been corrected (Line 398-399; Line 421; Line 425-428).

Supplementary material

1.4 RT reaction: please write cDNA and not CDNA.

My Response: The CDNA in the sentence has been modified to cDNA (supplementary materials: 1.4 RT reaction)

Page 4 – Please explain the gel electrophoresis results on page 4. The legend of the figure indicates the detection of dsRNA, but in the figure is seen VgR and Vg4.

My Response: The expression of the sentence has been modified, the modified sentence: Result analysis: DsVg4 and dsVgR were used as templates to clone Vg4 and VgR gene products with clear bands, consistent with the expected size (supplementary materials: page 4).

Round 3

Reviewer 1 Report

The text is now clear and I consider the manuscript ready for publication in IJMS after some minor corrections and improvements.

Line 106-106 - Revise text of the two sentences in lines 104-106.

Line 121 – species name Murraya odorifera should be in italics.

Figure S3 and S4 – There is no need to show figures S3 and Figure S4 as results are already visible in Figure S2.

Line 210 -  Interference groups are named Vg4 or VgR at the initial part of the text and from line 210 onwards they are named dsVg4 and dsVgR. Please use always the same nomenclature.

Table 2 -  Please complete the sentence in Table 2 caption: Comparison of D. citri oviposition and nymphs…Should it be:  Comparison of D. citri oviposition and nymphs formation?

Line 263 – the word “methods” is repeated.

Line 293 - Trichomonas vaginalis is not in italics.

Lines 374-375 -  Names of genes are usually written in lower case, please verify.

Line 412 – Eppendorf abbreviation was recorded as ep, however the name Eppendorf is written again and the abbreviation appears in upper case letters as EP (line 440 and 501). Please uniformize this information.

Line 549 – The word “contrast” is used as synonym of “control”. Please write control instead of contrast. Revise also Figure S2 where the word contrast is used.

Author Response

Response to Reviewer 1 Comments

Thank you for your valuable comments on our article. According to your suggestions, we have corrected again several points addressed by reviewers. The detailed point-by-point responses were listed below.

The text is now clear and I consider the manuscript ready for publication in IJMS after some minor corrections and improvements.

My Response: Thank you for your suggestions.

Line 106-106 - Revise text of the two sentences in lines 104-106.

My Response: The sentence has been modified, and the modified sentence: Indirect delivery of dsRNA to D. citri through plants was less reported and the observation time was usually short, longest observation time was not exceeding 25 days [22]. The effects of dsRNA long-term (more than 30 days) feeding on gene interference expression of D. citri are almost unknown (Line 85-88).

Line 121 – species name Murraya odorifera should be in italics.

My Response: Done (Line 100).

Figure S3 and S4 – There is no need to show figures S3 and Figure S4 as results are already visible in Figure S2.

My Response: Figure S3 and Figure S4 in the supplementary materials have been deleted (supplementary materials).

Line 210 -  Interference groups are named Vg4 or VgR at the initial part of the text and from line 210 onwards they are named dsVg4 and dsVgR. Please use always the same nomenclature.

My Response: The interference group has been uniformly named Vg4 interference group and VgR interference group (Line 183-501).

Table 2 -  Please complete the sentence in Table 2 caption: Comparison of D. citri oviposition and nymphs…Should it be:  Comparison of D. citri oviposition and nymphs formation?

My Response: The sentence has been modified, and the modified sentence: Comparison of D. citri oviposition and nymphs formation in interference group and control group (Line 224).

Line 263 – the word “methods” is repeated.

My Response: Done (Line 231).

Line 293 - Trichomonas vaginalis is not in italics.

My Response: Done (Line 256).

Lines 374-375 -  Names of genes are usually written in lower case, please verify.

My Response: The gene name has been changed to an abbreviation (Line 327-328).

Line 412 – Eppendorf abbreviation was recorded as ep, however the name Eppendorf is written again and the abbreviation appears in upper case letters as EP (line 440 and 501). Please uniformize this information.

My Response: The problematic abbreviation has been modified (Line 385; Line Line 388; Line 416; Line 445).

Line 549 – The word “contrast” is used as synonym of “control”. Please write control instead of contrast. Revise also Figure S2 where the word contrast is used.

My Response: Contrast has been replaced with control, Figure S2 has been modified (Line 491; supplementary materials).